

# Friction torque of wind-turbine pitch bearings – comparison of experimental results with available models

Matthias Stammler[1], Fabian Schwack[2], Norbert Bader[2], Andreas Reuter[1], Gerhard Poll[2]

[1] Fraunhofer IWES, Appelstraße 9A, 30167 Hanover, Germany
[2] IMKT, Leibniz Universität Hannover, Welfengarten 1 A, 30167 Hanover, Germany

*Correspondence to*: Matthias Stammler (matthias.stammler@iwes.fraunhofer.de)

**Abstract.** Pitch bearings of wind turbines are large, grease-lubricated rolling bearings that connect the rotor blades with the rotor hub. They are used to turn the rotor blades to control the power output and/or structural loads of the turbine. Common actuators turning the blades are hydraulic cylinders or electrical motor / gearbox combinations. In order to design pitch actuator systems that are able to turn the blades reliably without imposing an excessive power demand, it is necessary to predict the friction torque of pitch bearings for different operating conditions. In this paper, the results of torque measurements under load are presented and compared to results obtained using different calculation models. The results of this comparison indicate the various sources of friction that should be taken into account for a reliable calculation model.

## 1 Introduction

Pitch bearings (also called blade bearings) are subject to unfavorable operating conditions as they have to accommodate high bending moments while stationary or rotating at very low speeds. The connected parts, especially the rotor blades, provide limited stiffness. Usually, four-point contact ball bearings are used for this application, but for newer models of turbines, three-row bearing types have been chosen as well (Stammler and Reuter, 2015; Burton, 2011).

Pitch bearings are driven by combinations of electric motors and gearboxes with a total ratio exceeding 1:1000, or by hydraulic systems. In order to guarantee emergency-stop capability, accumulators have to store and provide sufficient energy for at least one pitch movement into the feather position should serious faults occur in the pitch system (Burton, 2011).

In order to design pitch actuator systems that are able to turn the bearings reliably but do not require excessive power, it is necessary to predict the friction torque for pitch bearings under all operating conditions.

Several equations and numerical models are available to calculate the friction torque of rolling bearings. However, there are no publications which compare them with experimental results of pitch bearings. In this paper, experimental results obtained at the Fraunhofer IWES pitch bearing test rig in Bremerhaven are compared to the results of different calculation models. The models considered range from two bearing manufacturers' catalog equations (SKF, 2014; Rothe Erde, 2016), which are based on PALMGREN's classical approach for friction prediction (Palmgren, 1957), to the numerical model developed by WANG (Wang, 2015).




## 2 Methods

### 2.1 Test rig, torque measurement and finite element model

The pitch bearing test rig at Fraunhofer IWES (see Figure 1) is designed for bearings of 3 MW-class wind turbines. In order to reproduce the operating conditions of pitch bearings, all interfaces (hub, blade, pitch drive) are the same as in the actual

wind turbine.

All loads are applied by hydraulic cylinders which are connected to ropes (see red rectangle in Figure 1). The loads are measured by means of load cells. The rope is attached to a load frame, whose center point is 30 m from the blade root.

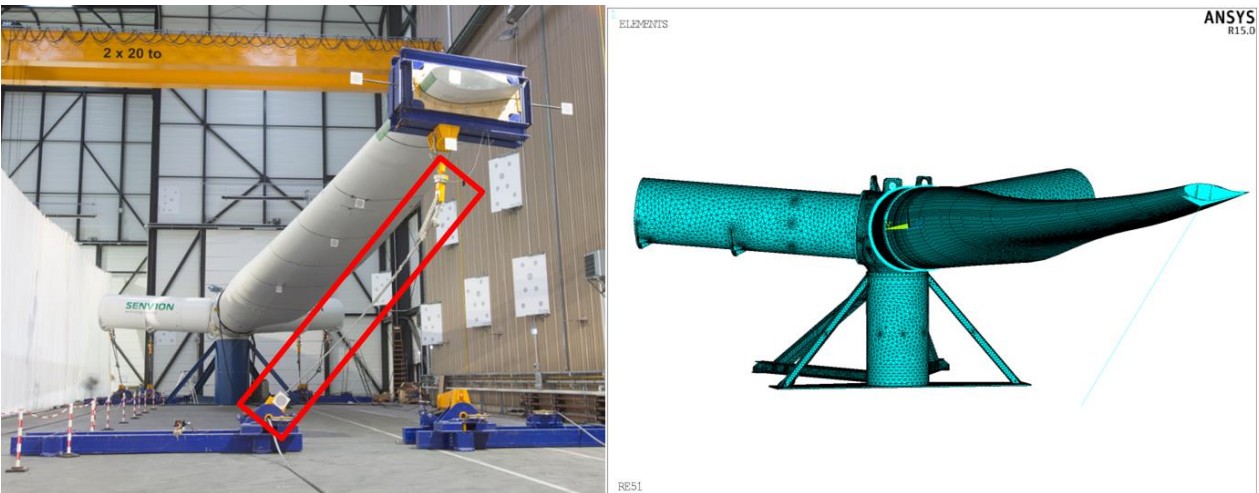

**Figure 1: Pitch bearing test rig at Fraunhofer IWES and corresponding FE model.**

The bending moment applied to the bearing is calculated with the force measured and the load vector, which is calculated with the aid of an optical measurement system. This optical measurement system consists of four cameras and several reflecting marks. Some of these marks are reference marks with known positions. These references are used to calculate the

position of the camera and the coordinates of the marks of interest. At the rotor blade, three marks indicate the current deflection (see Figure 2). Another mark is fixed to the lower end of the load rope.



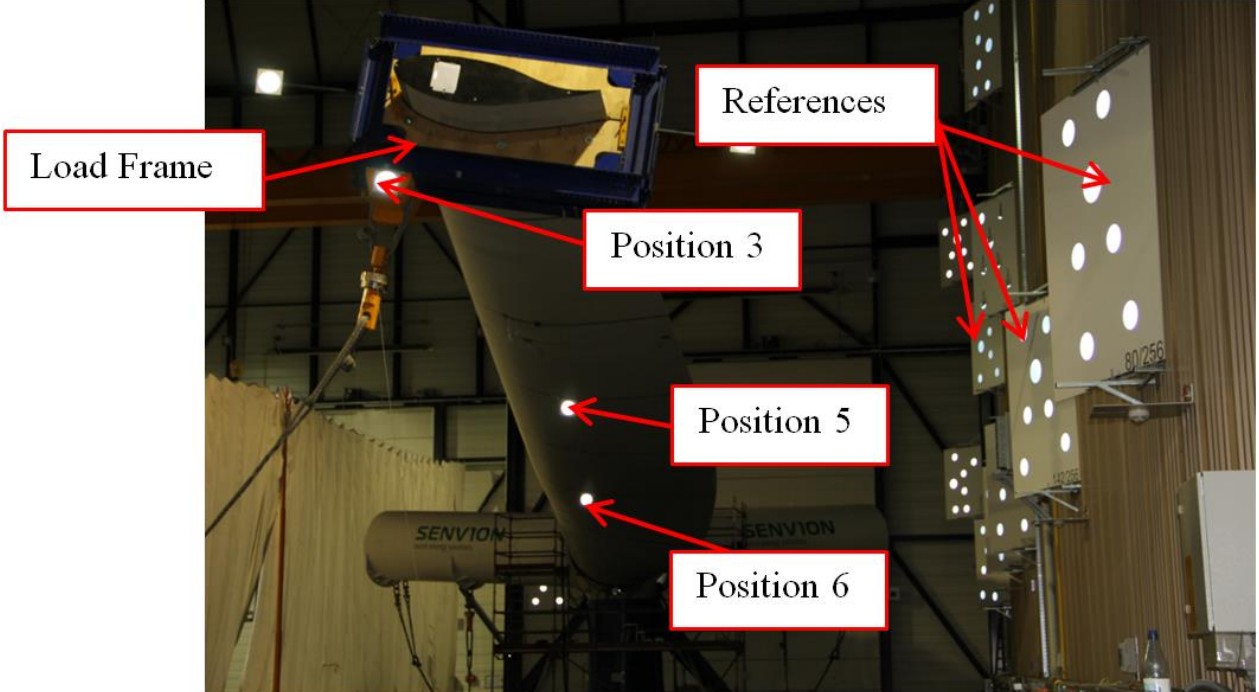

**Figure 2: Optical position measurement.**

5    The pitch drive is equipped with a strain-gauge torque measurement system at the pinion shaft on the low-speed side. The measurement system has been calibrated on the basis of known external loads.

The bearing is a grease-lubricated two-row four-point bearing of a 3MW-class turbine with an outer diameter of roughly 2.3 m. In addition to the tests with a mounted rotor blade, tests without the blade were executed to obtain data for zero load. The torque measurements were carried out under different pitch speeds and different external forces. The oscillating movements

10   of the bearings had a peak-to-peak amplitude of 10°. The torque values were measured for the middle 5° of these movements (see "Cleaned torque" curve in Figure 3).



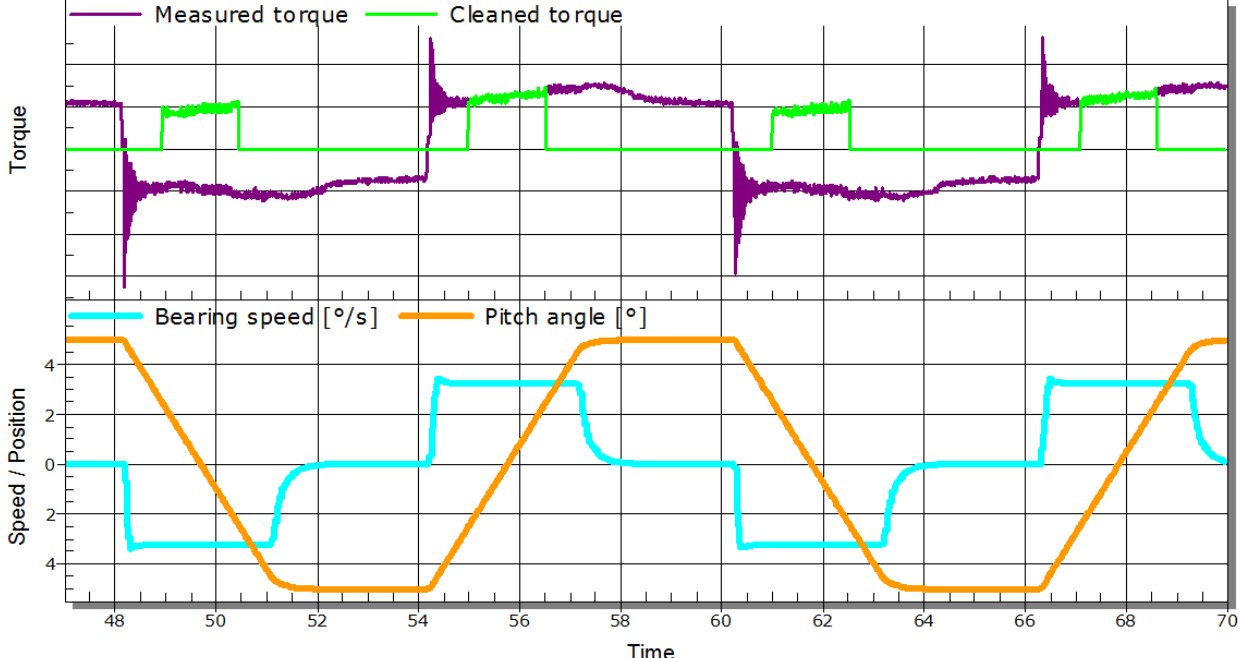

**Figure 3: Oscillating movements and torque measurements (example)**

To determine the load on the rolling elements of the bearing, a finite element (FE) model of the test rig was set up in ANSYS

15. These element loads are necessary for some of the friction calculation models.

### 2.2 Bearing friction issues

Simple bearing friction models use external influences (speed and load) to calculate the overall friction of a bearing. To obtain sound results with such an approach, it is necessary to actually measure the bearing friction under different external conditions. A change to the bearing system, e.g. a different lubricant or sealing, will require a new measurement to determine

the bearing friction in order to obtain exact results.

To actually predict the friction behavior of a bearing without the need for measurements, several friction mechanisms must be taken into account (Harris and Kotzalas, 2007). These mechanisms may be categorized according to influencing factors. For a given bearing, load, speed, or both influence the friction torque exerted by the different mechanisms:

• Load and speed dependent:
   o Heathcote (conformity) microslip due to differential velocities
   o Sliding due to roller body spinning
   o Rolling friction due to lubricant movements in the rolling contact





       o   Sliding of rolling bodies against cage / spacer

       o   Cage sliding against bearing rings

- Load dependent:

       o   Sub-surface hysteresis due to load changes

       o   Sliding of the sealing(s) against the bearing rings

- Speed dependent:

       o   Lubricant flow (churning) losses

In the following sections, two simple models (see Sections 2.3 and 2.6) and two more detailed models (Sections 2.4 and 2.5) are presented. In the literature, another model explicitly identified for blade bearing friction evaluation was evaluated (González et al., 2008), but as this model is contained within the more detailed model described in Section 2.5, it was not used for the subsequent calculations.

## 2.3 Palmgren's friction torque calculation model

In PALMGREN's model (Palmgren, 1957), the friction torque of a bearing is divided into a load-independent and a load-dependent part. The load-independent part $T_0$ takes into account an empirical value $f_0$ and the bearing diameter $D_M$, cf. Equation (1). A speed dependence is not part of the model for low rotational speeds as observed in pitch bearing applications. At higher rotational speeds, the lubricant viscosity $v$ and the bearing speed $n$ must also be taken into account.

$$T_0 = f_0 \cdot 10^{-7} \cdot 160 \cdot D_M^3 \ ; \text{if } (v \cdot n) < 2000 \ \frac{m^2}{60 \cdot s^2} \tag{1}$$

The load-dependent part $T_1$ depends on another empirical value $f_1$, the bearing diameter $D_M$ and the equivalent load $P_1$:

$$T_1 = f_1 \cdot P_1 \cdot D_M \tag{2}$$

The equivalent load $P_1$ is defined as the sum of the absolute values of all individual ball loads derived from the FE calculations described in Section 2.1. Note that according to the PALMGREN model for the low-speed regime, the friction torque is independent of the rotational speed of the bearing.

As the empirical values are not available for the pitch bearing used for this test, a minimum square deviation over all measured points is used to best fit the results obtained. Since the parameters have to be fitted to experimental data, the model is of little use to predict the friction of untested types of bearing.



### 2.4 Bearing manufacturer's friction model 1

The PALMGREN model was further refined to take account of the various components of the bearing that contribute to the total friction. One of these models was developed by a bearing manufacturer (SKF, 2014). The following Equation (3) shows the different elements of the total friction $M_{tot,S}$ used in this model:

$$M_{tot,S} = M_{rr} + M_{sl} + M_{seal} + M_{drag} \, , \qquad (3)$$

where $M_{rr}$ is the friction caused by rolling and $M_{sl}$ the friction caused by sliding movements, $M_{seal}$ the friction caused by the sealing and $M_{drag}$ the drag caused by the lubricant flow. $M_{rr}$ is calculated as follows:

$$M_{rr} = \phi_{ish} \cdot \phi_{rs} \cdot G_{rr} \cdot (v \cdot n)^{0,6} \, , \qquad (4)$$

where $\phi_{ish}$ and $\phi_{rs}$ are factors to take account of the lubricant film thickness and the lubrication displacement, and $G_{rr}$ is a base value for the rolling friction. This value depends on external loads.

To calculate $M_{sl}$, a coefficient for the sliding friction, $\mu_{sl}$, and a base value for the sliding friction, $G_{sl}$, are multiplied together, cf. Equation (5):

$$M_{sl} = \mu_{sl} \cdot G_{sl} \, , \qquad (5)$$

The sealing friction $M_{seal}$ is calculated according to Equation (6):

$$M_{seal} = K_{S1} \cdot d_s^{\beta} + K_{S2} \, , \qquad (6)$$

where all values are empirically determined and depend on the bearing type, sealing type and the bearing diameter. These
values are not available for pitch bearings and are chosen so as to deliver a best fit for the measured values of the overall friction torque at zero load.

The last part of $M_{ges}$ is the friction caused by lubricant flow, $M_{drag}$. Equation (7) contains the individual elements used to calculate this component. $V_M$ is a factor determined by the lubricant's resistance against movement, $K_{Ball}$ a factor taking into



account the behavior and number of the rolling elements, $f_t$ a factor taking into account the amount of lubricant in the bearing, and $R_S$ a value depending on the bearing type.

$$M_{drag} = 0.4 \cdot V_M \cdot K_{Ball} \cdot D_m^5 \cdot n^2 + 1.093 \cdot 10^{-7} \cdot n^2 \cdot D_m^3 \cdot \left(\frac{n \cdot D_m^2 \cdot f_t}{v}\right)^{-1.379} \cdot R_S \qquad (7)$$

## 2.5 DING WANG rheological model

WANG (Wang, 2015) considers enhanced rheological fluid models and experimental results in his model for calculating the friction torque. The model follows the assumptions of STEINERT (Steinert, 1995) and ZHOU (Zhou and Hoeprich, 1991), which divide the friction torque into different parts which can be calculated independently from each other. For ball bearings, these parts are the friction torque, which results from the irreversible deformation work on the bearing steel $M_{def}$, the torque from the hydrodynamic rolling friction $M_{roll}$ and, if the bearing is axially loaded and the contact angle $\alpha$ is greater than 0, the torque caused by the spinning friction $M_{spin}$ (FVA, 2010). The friction moment $M_{HC}$ takes account of the differential slippage which occurs due to the different velocities in the contact between ball and raceway, also known as the HEATHCOTE effect (Harris and Kotzalas, 2007).

Equation (8) shows the different parts in a mathematical relationship for a better understanding. The parts $M_{spin}$ and $M_{HC}$ cannot be analyzed separately because they both occur in the sliding moment $M_{slide}$, see Equation (9).

$$M_{ges,W} = M_{def} + M_{roll} + M_{spin} + M_{HC} \qquad (8)$$

$$M_{slide} = M_{spin} + M_{HC} \qquad (9)$$

The friction moment of the deformation work $M_{def}$ based on the approach of JOHNSON (Johnson, 1970) considers the damping of the material $\kappa$, the semi-axis of the Hertzian contact ellipse $b$, and the load on each roller $Q$:

$$M_{def} = \frac{3}{16} \cdot \kappa \cdot b \cdot Q \qquad (10)$$

To calculate $M_{roll}$ the energy balance between the subsystem and the whole system needs to be considered (Steinert, 1995):

$$M_{roll} = \frac{D_{RE}}{2} \sum_{i=1}^{Z} \left[ \left| \frac{\omega_{RE}}{\omega_{IR}} \right| \left( F_{roll,IR,i} + F_{roll,OR,i} \right) \right] \qquad (11)$$



Equation (11) takes into account the diameter of the rolling element $D_{RE}$, the angular velocity of the rolling elements $\omega_{RE}$ and the inner ring $\omega_{IR}$, and the hydrodynamic forces at the inner $F_{roll,IR,i}$ and outer ring $F_{roll,OR,i}$ of each rolling element $i$.

According to ZHOU (Zhou and Hoeprich, 1991), the hydrodynamic force $F_{roll}$ can be calculated from the isothermal hydrodynamic force $F_{roll,isoth}$ (Goksem and Hargreaves, 1978), and from a factor $C_{th}$ which takes account of losses due to shear and compression heating to obtain a linear relationship between film thickness and hydrodynamic rolling friction (Baly, 2005).

$$F_{roll} = C_{th} \cdot F_{roll,isoth} \tag{12}$$

As mentioned before, the physical effect which leads to a sliding moment $M_{slide}$ cannot be calculated separately. In the numerical model, $M_{slide}$ can be calculated due to the relative sliding velocities and the resulting local shear stresses. For a better understanding, the physical effects in the sliding moment will be briefly explained.

$M_{spin}$ is calculated from the inner diameter $r$, the contact area $A$, and the shear stress $\tau$ in the lubricant, which can be
determined with the aid of the Newtonian shear stress approach (FVA, 2005):

$$M_{spin} = \int_A r \cdot \tau \cdot dA \tag{13}$$

It must be borne in mind that it is difficult to apply this approach for $M_{spin}$ because most lubricants exhibit non-linear flow behavior. These non-Newtonian lubricants exhibit shear thinning and a limiting shear stress under elastohydrodynamic lubrication (EHL) conditions. $\tau$ needs to be limited to $\tau_{lim}$ due to rheological effects (Wang, 2015).

The friction from the differential slippage / HEATHCOTE effect depends on the energy balance and takes into account the frictional losses at the inner ($P_{HC,IR,i}$) and the outer ring ($P_{HC,AR,i}$) of each rolling element $i$:

$$M_{HC} = \sum_{i=1}^{z} \left[ \frac{1}{\omega_{IR}} \left( P_{HC,IR,i} + P_{HC,AR,i} \right) \right] \tag{14}$$

All sliding losses in the contact lead to high shear rates within the contact areas and are thus dominated by the lubricant behavior. Empirical results are used to calculate the friction torque which takes into account the limiting shear stress. These results are obtained using a two-disc machine (see Figure 4).





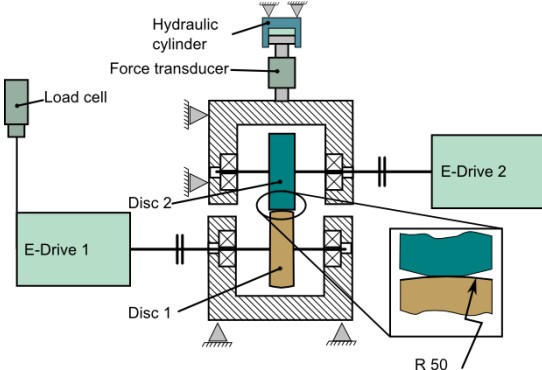

**Figure 4: Two disc machine**

The relationship between the medium shear stress $\bar{\tau}$ and the shear rate $\dot{\gamma}$ in the EHL contact is important to take account of rheological effects (Poll, 2011; Poll and Wang, 2012). A relationship between maximum shear stress $\tau_{max}$ and the mean pressure $p$, temperature $T$, and the shear rate $\dot{\gamma}$ can be obtained from the experiment. If the calculated Newtonian shear stress $\tau$ is greater than the maximum shear stress $\tau_{max}$ from the experiment, the shear stress is truncated at the maximum value. The local shear stress is then integrated to yield the total friction force of the contact. The sum of the contact friction losses

gives the friction torque due to sliding (Leonhardt et al., 2016).

**2.6 Bearing manufacturer's friction model 2**

While the two aforementioned approaches try to split the friction torque according to different friction causes, the following model is taken from a manufacturer's current bearing catalog (Rothe Erde, 2016). This method has no speed-dependent component and is adapted to different bearing types by the friction coefficient $\mu$. The friction torque $M_r$ is calculated

according to Equation (15):

$$M_r = \frac{\mu}{2} \cdot (4.4 \cdot M_k + F_A \cdot D_M + 3.81 \cdot F_R \cdot D_M) \tag{15}$$

This model does not take into account any load-independent part. In practice, however, all bearings experience frictional losses even under unloaded conditions. Consequently, the bearing manufacturer states that the equation must not be used for

unloaded conditions.



## 3 Results & Discussion

### 3.1 Validation of the FE model

The results of the FE calculations have been compared to the results of the optical measurement system described in Section 2.1. A bending moment of 5 MNm is applied both to the real test rig and in the finite element (FE) model, and the deformations at the three positions are measured and compared with the model results. Figure 5 shows the deviations of the total deformations between the FE calculation and test rig measurements.

| Position | Deviation [%] |
|----------|---------------|
| 6        | 26.4          |
| 5        | -1.5          |
| 3        | 3.3           |

**Figure 5: Deviations of deformations between FE model and test rig measurements**

While positions 5 and 3 show satisfactory agreement, position 6 shows a large relative deviation between FE analysis and test rig. The large relative deviation is partly caused by the low absolute deformation (less than 20 mm) near the blade root (small absolute values result in high relative values) and the longer distance to the camera positions, which results in higher uncertainty.

### 3.2 Friction torque measurements and comparison with models

Figure 6 shows the results of friction torque measurements at different rotating speeds of the pitch bearing. The measurements were executed in steps of 1 MNm, all other values are interpolations between the measurements. The external load was applied via a load frame (see Figure 2) and is expressed as the resulting bending moment at the blade root. The measurements for each load-speed-combination were repeated at least 20 times with no significant deviation between the mean values of friction torque. However, owing to the oscillating movements used for the torque measurements, there is a relatively high standard deviation in the single measurements (shown for 2 and 5 MNm in Figure 6).




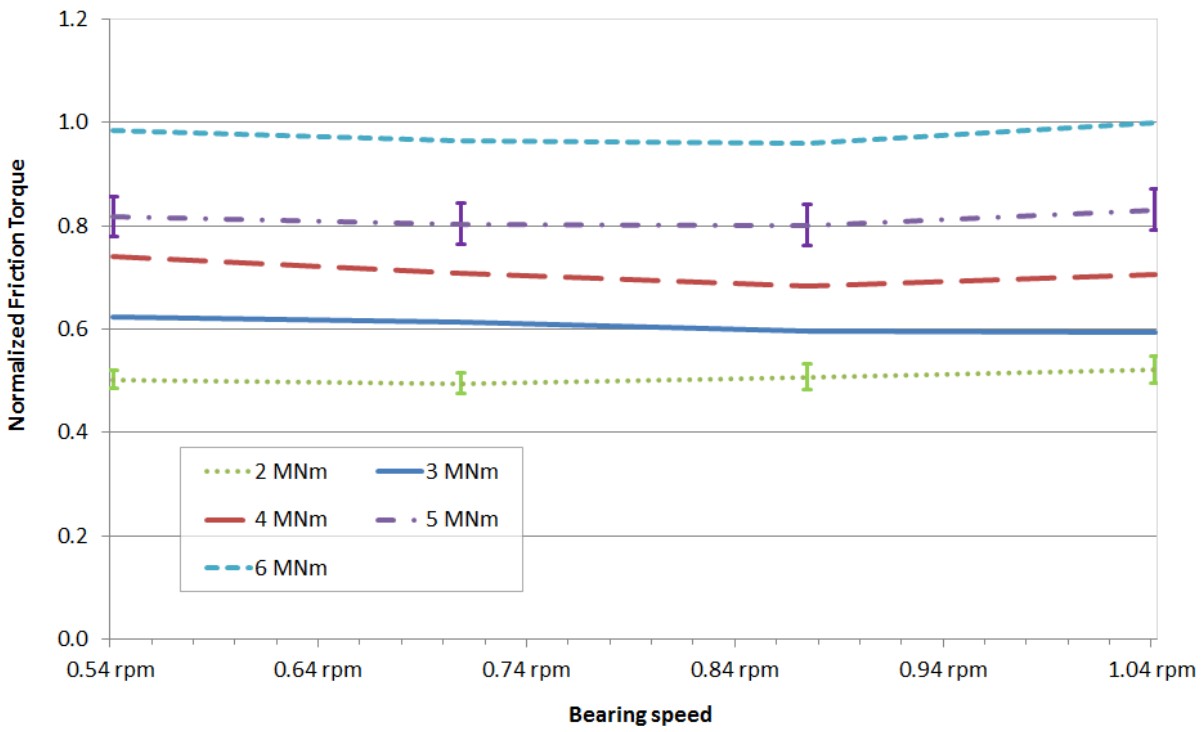

**Figure 6: Measured friction torque at different speeds and loads**

The values of the friction torque are normalized to the highest value of the measurements. For the conditions shown, the

5    theoretical lubricant film thickness according to Dowson and Hamrock (Dowson and Hamrock, 1977) is close to or above the combined surface roughness of raceway and roller (Stammler and Poll, 2014). As the bearing is grease-lubricated, lubricant starvation might further reduce the lubrication film thickness, thus a mixed lubrication regime is the most likely lubrication condition. The speeds tested are within the usual range of pitch bearing speeds. From the measured values, it is not possible to derive the speed dependence of the friction torque.

In Figure 7, the values are again normalized to the highest friction torque measured. The error bars refer to the standard deviation of the measured values. For the first manufacturer's and the PALMGREN calculations, previously unavailable empirical values had to been chosen to match the curves with the measured values. For the PALMGREN calculation, the value $f_0$ was adapted for the zero load condition and the value $f_1$ was adapted in order to make the difference between zero

15    load torque and the highest load torque match the measured values. Currently, the PALMGREN model cannot be used to predict the friction torque of other pitch bearings as there are no available values for the empirical factors $f_0$ and $f_1$. It is unclear whether the values used in this work are correct for loads higher than the measured loads or other bearing



diameters. $f_1$ was adapted to match the slope between 0 and 6 MNm external load; if it had been adapted to the slope between 2 and 4 MNm, the differences between measurements and model calculations would have been higher.



### Friction torque at different loads (averaged over speeds)

**Figure 7: Measured and calculated friction torque at different loads**

The first manufacturer's model needed additional adjustment of the sealing friction to match the zero load torque. The empirical values provided with the model do not include values for the sealing friction of bearings with diameters comparable to typical pitch bearing diameters. Sealing friction is given for bearings with a maximum diameter of 340 mm.

10  Thus, the $K_{S1}$ value that is part of the sealing friction was set in such a way that the non-load friction matches the measured values. In order to achieve a match, the value had to be raised drastically. Comparing the individual elements of the model, the adjusted $M_{seal}$ is by far the largest part of the calculated friction at zero load and makes up nearly 99% of the friction at 2 MNm, which does not seem plausible. Additionally, the load dependence of the friction torque is underestimated by 67 % in comparison to the measurements. This may be caused by the $M_{seal}$ part as well, as a four-point bearing suffers relatively

15  large deformations of the bearing rings under loads and should exhibit a load-dependent behavior of the sealing friction.
The second manufacturer's equation is explicitly not intended for zero load; as such this value is not displayed in the chart. The friction torque calculated with this model deviates by 35% from the measured values at a load of 2 MNm, and by 10% at





a load of 6 MNm. As the slope of the load dependence of the calculated curve is 15% higher than that of the measured values, it might result in overestimated friction torques at loads higher than 6 MNm.

The model proposed by WANG does not take account of the sealing friction and shows the friction torque to have a rather high speed dependence that does not match the measured values. The model was originally intended for the calculation of

5    friction under fully lubricated conditions and needs some further adjustments for mixed friction conditions.

Figure 8 shows the speed dependence of the different calculation methods, and the measurement results at 5 MNm external load. While the measurement values and most of the model calculations show no speed dependence, the model of WANG contains a dependence on bearing speed.

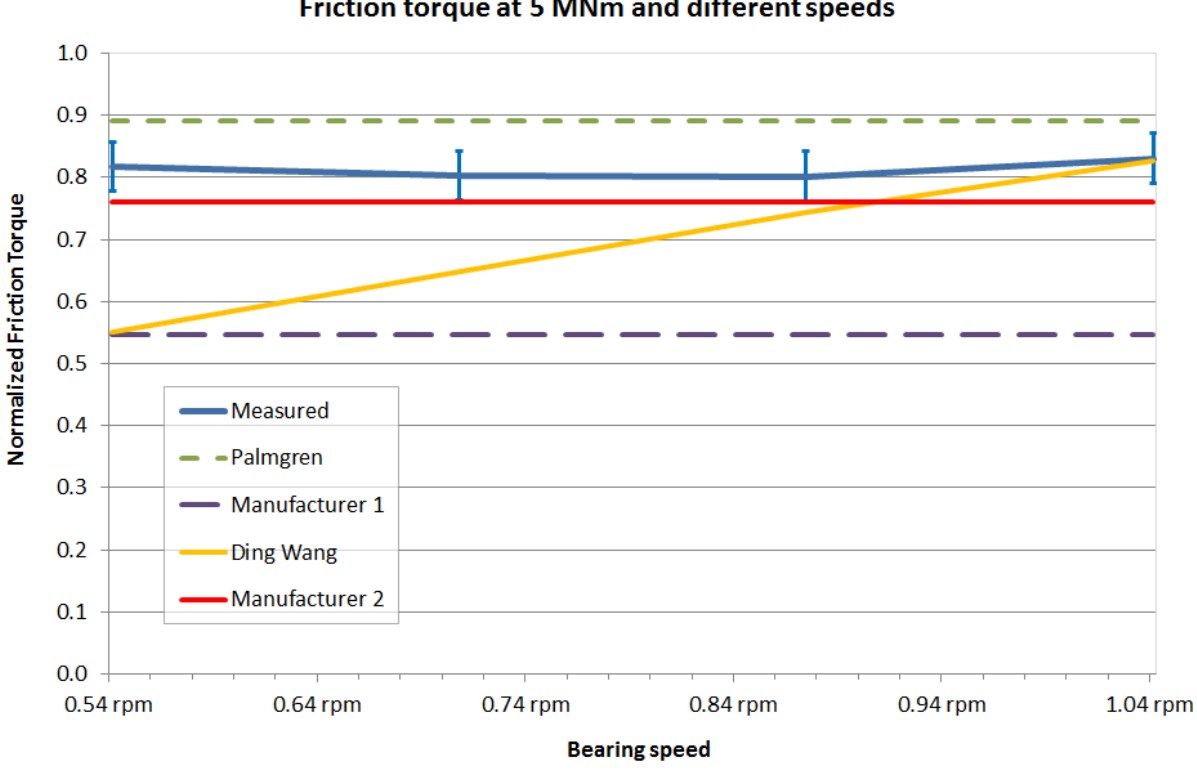

**Figure 8: Measured and calculated friction torque at different speeds and 5 MNm load**

Only the models from WANG and from the second manufacturer can be used to predict the friction torque of pitch bearings

15    without additional tests, as the other models need the adjustment of empirical values. The second manufacturer's model relies on empirical values as well, but in this case these values for different types of pitch bearings are available in the public domain.





| Model | All empirical values available | Speed behavior compared to measurement | Load behavior compared to measurement | No-load friction compared to measurement |
|---|---|---|---|---|
| **PALMGREN** | **No** | **Good** (no speed dependence) | - (Adjusted to empirical value) | - (Adjusted to empirical value) |
| **MANUFACTURER 1** | **No** | **Good** (no speed dependence) | **Poor** (67 % underestimated) | - (Adjusted to empirical value) |
| **WANG** | **Yes** | **Poor** (55% average increase from 3 to 6 °/s) | **Poor** (39 % underestimated) | **Not determined** |
| **MANUFACTURER 2** | **Yes** | **Good** (no speed dependence) | **Fair** (15% overestimated) | - |

**Figure 9: Overview of agreement between bearing friction models and experimental results**

## 3.3 Conclusions

In this paper, torque measurements of a loaded four-point ball-type pitch bearing on which realistic interfaces were mounted, have been presented. The measurements were executed at a pitch bearing test rig with realistic interfaces (hub, pitch actuator, blade). While the measurements show a clear load dependence, no systematic dependence on the rotational speed of the bearing is observed within the range of speeds tested.

The load dependence exhibits nearly linear behavior, with a positive value at zero load condition. This supports the assumption that the friction torque has a load-independent part that is present in all of the calculation methods except for the second manufacturer's model.

With the PALMGREN model, empirical values were adapted to match the measurement results, but it is doubtful if these values match other load conditions and / or types of pitch bearing. The sealing friction part of the first manufacturer's model was adjusted to match the measured values at zero load. This led to a dominance of the sealing friction, which does not seem plausible. As such, it may be concluded that the other parts of the friction are underestimated by this model. The second manufacturer's model overestimates the load dependence of the friction. The WANG model overestimates the speed dependence of the total friction.

None of the models reviewed is able to predict all aspects of the friction torque behavior of the pitch bearing. With the PALMGREN, WANG and first manufacturer's models, this may be due to the range of bearings taken into account to create the models. Both the bearing types and sizes underlying the models differ significantly from those of typical pitch bearings. Additionally, it can be assumed that most of the experiments leading to the creation of these models were conducted under close-to-ideal lubrication conditions with oil lubrication, fully flooded contacts and a complete separation between raceway





and rollers. In grease-lubricated pitch bearings, mixed lubrication is possible under normal operating conditions. As such, the results have only limited comparability to the models based on tests under better lubrication conditions.

With none of the models being able to reliably predict the friction torque of the pitch bearing in the test described, the only way to currently determine the friction torque is with the aid of measurements. In future work, the test rig will be used for

further friction torque measurements with different bearings to support the development of models suitable for large grease-lubricated bearings like pitch bearings. Further development work on the models will take into account the influence of the sealing, the lubrication conditions within pitch bearings, and the characteristics of different types of pitch bearing.

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
