# Peer review of "Friction torque of wind-turbine pitch bearings – comparison of experimental results with available models"

_Wind Energy Science, 2017_

## Author Comment (AC1) · 10 Jun 2017

Thank you very much for your comments! In the following lines, we would like to comment on the single aspects, using the original critics as a starting point. A revision of the paper has been created and some of the following remarks refer to this revision:

- "The friction torque measurement system is described very briefly (line 5, page 3), despite its central importance for the presented work. There should be at least a figure showing conceptually the configuration of the pitch drive and the instrumentation . Corresponding text describing such a figure should also be added."

Measurement of shaft torques is a standard procedure in many industries, as such we did not deem it worthwile to get into detail here. Nevertheless, we added some additional information about the telemetry system on page 5.

- "Too little information regarding the application of the different friction models is given, in order to support the conclusion that all considered models are insufficient. The information given from line 10 on page 11 to the end of page 13 should be more thorough and be presented more systematically."

We have reordered the information in the following way: At first, the adaptions of empirical values are described. Second, the results oft he single models are explained. We think this is more systematically than before, thanks for the remark. We also added some further explanations on single aspects oft he results.

- "It is also unclear how the roller element loads extracted from the FE model are used here. An example of a corresponding FE result would help here."

The FE consists in the load sharing of the individual rolling elements. Such calculations have been described in the literature and respective references have been added on page 4. The rolling elements load are used to calculate the overall load used for some of the models or for some of the calculations necessary for the Wang model.

- "Normalizing with respect to the average torque measured at the maximum load, limits the scientific value of the presented data significantly. Since the measurements appear to be the main contribution of the model, either the absolute values should be provided, or data normalized with respect to a well defined quantity."

We did not normalize against average values, but against the highest measured value. A clarification has been added on page 11.

- "The term movement is in general too broad for a scientific text. In many places of the text the term can be replaced with more specific ones such as rotation and shearing."

Indeed, rotation is a more suitable expression in many of the cases. This has been

changed.

- "In several places the subscript "ges" is used instead of the English equivalent "tot"."

The initial idea was to maintain the original subscripts in order to make it easier to compare the equations mentioned in the paper against the ones in the references. But we understand the subscript "tot" is self-explanatory in an English paper and exchanged "ges" against "tot" in the remaining instances.

- "the argument in line 19 on page 10 is unclear" The argument was completed by an explanation: However, owing to the oscillating movements used for the torque measurements, there is a relatively high standard deviation in the single measurements (shown for 2 and 5 MNm in Figure 6), due to torque vibrations caused by the repeated accelerations of the blade and pitch bearing masses.

- ""had to be chosen" (line 13, page 11)"

Rectified.

A new version of the manuscript is prepared and added to this comment. It will be uploaded as a new manuscript version according to the editor's decision.

Please also note the supplement to this comment:
http://www.wind-energ-sci-discuss.net/wes-2017-20/wes-2017-20-AC1-supplement.pdf

**Supplement:**

[revised manuscript text omitted]

---

## Referee Comment (RC2) · Anonymous Referee #2 · 23 Oct 2017

When they compare the measured friction torque value to the calculated values with four models, authors should give the parameter values and select reasons for these models.

---

## Author Response (AR1)

**Friction torque of wind-turbine pitch bearings – comparison of experimental results with available models**

Matthias Stammler[1], Fabian Schwack[2], Norbert Bader[2], Andreas Reuter[1], Gerhard Poll[2]

[1] Fraunhofer IWES, Appelstraße 9A, 30167 Hanover, Germany
[2] IMKT, Leibniz Universität Hannover, Welfengarten 1 A, 30167 Hanover, Germany

*Correspondence to*: Matthias Stammler (matthias.stammler@iwes.fraunhofer.de)

**Consolidated reply to review comments**

**1 Description**

All reviewer comments are in *italic* letters, the reply of the authors is given directly below each comment. Text taken from the revised paper is in **grey bold** letters. The comments or put into sections according to topics / types of the comments. The following comments are all comments given by the reviewers. Each comment has been answered and several changes have been made to the manuscript.

**2 Torque measurement system**

*The friction torque measurement system is described very briefly (line 5, page 3),despite its central importance for the presented work. There should be at least a figure showing conceptually the configuration of the pitch drive and the instrumentation .Corresponding text describing such a figure should also be added.*

Torque measurement on shafts is a standard procedure for many applications and there are some commercial companies offering such systems. Torque measurements are available by the following companies (non-exclusive list):

*KTR, Datum Electronics, Cedrat Technologies*

Please refer especially to the following system, which was used in this test:
http://www.axon-systems.eu/products/1-channel-telemetry-systems-strain-gauge-temperature/axon-j1?lang=en

As these systems are widespread, a detailed explication in the paper seemed unnecessary. We did, however, add a few sentences on the principle, and a figure showing the orientation of the strain gauges on page 5:

**The pitch drive is equipped with a strain-gauge torque measurement system at the pinion shaft on the low-speed side. A full bridge of strain gauges is mounted on the shaft (see Figure 3), together with a rotary unit. Data transfer and power supply is done telemetrically via a ring stator. The measurement system has been calibrated by applying known torques to the shaft.**

[Figure]

Figure 3: Torque measurement strain gauges on shaft.

**3 Friction model results, calculation, normalization and discussion**

*Too little information regarding the application of the different friction models is given, in order to support the conclusion that all considered models are insufficient. The information given from line 10 on page 11 to the end of page 13 should be more thorough and be presented more systematically. It is also unclear how the roller element loads extracted from the FE model are used here. An example of a corresponding FE result would help here.*

*Normalizing with respect to the average torque measured at the maximum load, limits the scientific value of the presented data significantly. Since the measurements appear to be the main contribution of the model, either the absolute values should be provided, or data normalized with respect to a well defined quantity.*

*When they compare the measured friction torque value to the calculated values with four models,authors should give the parameter values and select reasons for these models.*

We have reordered the information in the following way: At first, the adaptions of empirical values are described. Second, the results of the single models are explained. We think this is more systematically than before, thanks for the remark. We also added some further explanations on single aspects of the results on pages 10, 11 and 12:

[revised manuscript text omitted]

The measurement data used for this paper was created during a commercial project for a customer. This data is subject to non-disclosure agreements. As such, absolute values cannot be given within this paper. For this very same reason, we did not include the individual values of empirical factors for the single models, as this would allow for the calculation of the absolute values.

We deem the scientific results worthy nonetheless: The validity of the friction models for large slewing bearings has not yet been evaluated in any publication known to the authors. The lack of appropriate models for friction torque calculation of blade bearings has been clearly shown and the need for further research in this field was stated. Due to this, there is also a lack of a "well-defined quantity" to normalize the values against – there is simply no comparable data available in the literature.

**3 Minor and practical comments**

*The term movement is in general too broad for a scientific text. In many places of the text the term can be replaced with more specific ones such as rotation and shearing.*

5   Indeed, rotation is a more suitable expression in many of the cases. This has been changed.

*In several places the subscript "ges" is used instead of the English equivalent "tot".*

The initial idea was to maintain the original subscripts in order to make it easier to compare the equations mentioned in the
10   paper against the ones in the references. But we understand the subscript "tot" is self-explanatory in an English paper and exchanged "ges" against "tot" in the remaining instances.

*the argument in line 19 on page 10 is unclear*

The argument was completed by an explanation:

15   **However, owing to the oscillating movements used for the torque measurements, there is a relatively high standard deviation in the single measurements (shown for 2 and 5 MNm in Figure 6), due to torque vibrations caused by the repeated accelerations of the blade and pitch bearing masses.**

*"had to be chosen" (line 13, page 11)*

20

Rectified.

---

## Author Response (AR2)

**Friction torque of wind-turbine pitch bearings – comparison of experimental results with available models**

Matthias Stammler[1], Fabian Schwack[2], Norbert Bader[2], Andreas Reuter[1], Gerhard Poll[2]

5  [1] Fraunhofer IWES, Appelstraße 9A, 30167 Hanover, Germany
[2] IMKT, Leibniz Universität Hannover, Welfengarten 1 A, 30167 Hanover, Germany

*Correspondence to*: Matthias Stammler (matthias.stammler@iwes.fraunhofer.de)

10  *Non-disclosure of absolute torque values*

In addition to the former discussion of single items, we would like to detail the reasons for not disclosing absolute values:

Blade bearings are customized, very large rolle rbearings designed for a specific wind turbine type. No manufacturer of
15  bearings offers a catalogue of blade bearings, as the requirements for the interfaces depend on the wind turbine design.
Further on, the adjacent parts (blade, hub, pitch drive) are specific to a wind turbine type as well. Tests of real blade bearings
are exclusively tests of bearings designed for a serial application. Intellectual property rights on bearings and interface parts
belong to the bearing manufacturer and/or the turbine OEM. Turbine OEMs in general are very sensitive towards data
disclosing. The environment for turbine manufacturers is highly competitive, and massive price reductions due to the auction
20  mechanisms further promote this situation. The tests which data we used to do the scientific analysis were done on a test rig
equipped with hub, blade and blade bearing by the company Senvion. Senvion did not agree in disclosing the absolute values,
as these allow using a more detailed friction model and in turn designing pitch drives better tailored to the actual requirements.
As such, this information is relevant for the competition and cannot be part of this part.
Yet, as we mentioned during the discussion, we deem the results of the work worthwhile nonetheless, as this is the first ever
25  publication regarding full-scale blade bearing friction torque. The comparison with the models gives a very good indication
on which model is suitable for practical implication.